# *Astragalus* Saponins, Astragaloside VII and Newly Synthesized Derivatives, Induce Dendritic Cell Maturation and T Cell Activation

**DOI:** 10.3390/vaccines11030495

**Published:** 2023-02-21

**Authors:** Nilgun Yakubogullari, Ali Cagir, Erdal Bedir, Duygu Sag

**Affiliations:** 1Department of Bioengineering, Izmir Institute of Technology, Izmir 35430, Turkey; 2Department of Chemistry, Izmir Institute of Technology, Izmir 35430, Turkey; 3Izmir Biomedicine and Genome Center, Izmir 35340, Turkey; 4Izmir International Biomedicine and Genome Institute, Dokuz Eylul University, Izmir 35340, Turkey; 5Department of Medical Biology, Faculty of Medicine, Dokuz Eylul University, Izmir 35340, Turkey

**Keywords:** vaccine adjuvant, triterpenoid saponin, semi-synthesis, immunomodulation, immunological evaluation

## Abstract

Astragaloside VII (AST VII), a triterpenic saponin isolated from *Astragalus* species, shows promise as a vaccine adjuvant, as it supported a balanced Th1/Th2 immune response in previous in vivo studies. However, the underlying mechanisms of its adjuvant activity have not been defined. Here, we investigated the impact of AST VII and its newly synthesized semi-synthetic analogs on human whole blood cells, as well as on mouse bone marrow-derived dendritic cells (BMDCs). Cells were stimulated with AST VII and its derivatives in the presence or absence of LPS or PMA/ionomycin and the secretion of cytokines and the expression of activation markers were analyzed using ELISA and flow cytometry, respectively. AST VII and its analogs increased the production of IL-1β in PMA/ionomycin-stimulated human whole blood cells. In LPS-treated mouse BMDCs, AST VII increased the production of IL-1β and IL-12, and the expression of MHC II, CD86, and CD80. In mixed leukocyte reaction, AST VII and derivatives increased the expression of the activation marker CD44 on mouse CD4^+^ and CD8^+^ T cells. In conclusion, AST VII and its derivatives strengthen pro-inflammatory responses and support dendritic cell maturation and T cell activation in vitro. Our results provide insights into the mechanisms of the adjuvant activities of AST VII and its analogs, which will be instrumental to improve their utility as a vaccine adjuvant.

## 1. Introduction

Vaccination is one of the best strategies to induce protective immunity against infectious diseases and cancer. Although discovering a safe and effective antigen is the main target of vaccinology, the development of an adjuvant has also been gaining equal importance in recent years. Adjuvants are synthetic or biological substances, and when they are administrated into the body, especially with an antigen giving a poor immune response, these compounds have the ability to increase the immunogenicity of the antigen with broad antibody and effective T cell responses [1,2].

For more than a century, few potent adjuvants with acceptable toxicity have been licensed to be used in human vaccines. While Alum (a mixture of compounds, mainly aluminum hydroxide and aluminum phosphate) [3,4,5,6] and oil-in-water emulsions containing squalene (MF59, AS03) are widely used, there has also been a focus on combining different immunomodulatory agents in a system to enhance the efficacy of vaccines. For example, Adjuvant System 01 (AS01), which contains monophosphoryl lipid A (MPLA) and the saponin natural product (QS-21), has been approved for GSK’s malaria (Mosquirix) and shingles (Shingrix) vaccines. Moreover, AS04, containing MPLA and alum, also has been approved for a hepatitis B vaccine (Fendrix) and human papillomavirus vaccine (Cervarix) [2,7]. For combinatory adjuvant system development studies, QS-21 was selected and has become the most widely studied saponin adjuvant for the past 25 years.

QS-21 is a triterpenic saponin isolated from *Quillaja saponaria* and has been investigated in clinical trials as a vaccine adjuvant for cancer, Alzheimer’s disease, malaria, HIV, etc. [8]. The compound demonstrates an IgG1/IgG2a balanced antigen-specific antibody response along with boosting IFN-γ and IL-2 and effective CD8^+^ T cell responses. The studies to reveal the action mechanisms of QS-21 demonstrated that this compound intercalates with cell membrane cholesterols resulting in pore formation, accelerates the antigen uptake by antigen-presenting cells, and induces the cross-presentation of antigen. In addition, the activation of NLPR3 inflammasome and further secretion of IL-1β and IL-18 had a role in the cross-presentation induced by QS-21 [9,10,11,12,13,14,15]. One of the promising semi-synthetic analogs of QS-21, GPI-0100, was developed by replacing the acyl group at the fucopyranosyl residue with dodecylamine-linked glucuronic acid (at C6) located at C3 of the aglycone and generated Th1 immunity and CTL response similar to QS-21 but with less toxicity [16]. Moreover, semi-synthesis studies on QS-21 were carried out in order to overcome the commercial development problems of QS-21, increase its efficacy in vaccine formulations, and reduce its toxicity [8,17,18,19,20]. Consequently, there is an urgent need to develop novel potent and less toxic adjuvants and to elucidate the cellular and molecular mechanisms of their activities.

*Astragalus* saponins have been widely used in Chinese Traditional Medicine for their pharmacological properties such as immunomodulatory, antioxidant, antitumor, and anti-viral properties [21]. Until today, there were few studies reporting the potency of *Astragalus* saponins to be used as a vaccine adjuvant. One of the cycloartane-type triterpenic saponins isolated from *Astragalus trojanus* [22] is Astragaloside VII (AST VII). In preliminary in vivo studies, we mainly focused on the investigation of the adjuvant potency of this compound. For that, AST VII was administered with various biological compounds and antigens (bovine serum albumin, lipopolysaccharide, Newcastle disease virus, influenza, etc.) and demonstrated a Th1/Th2 balanced immune response through antigen-specific IgG, IgG1, and IgG2b antibody responses and the production of IL-2, IFN-γ, TGF-β, and IL-17A [23,24,25,26,27]. In addition to inducing cellular and humoral immune responses, its high solubility in water, very low hemolytic activity at high concentrations, high stability, and suitability to lyophilization make AST VII a potent adjuvant candidate. However, its mechanism of action has not been defined yet.

In this study, we prepared new semi-synthetic derivatives of AST VII, namely DC-AST VII and DAC-AST VII, with the aim to find more potent compounds and shed light on the mechanisms contributing to their adjuvant actions. The cellular immune response obtained by the treatments of AST VII and its derivatives were investigated using innate immune cells and T cells. The findings provided here are the first evidence regarding the mechanism of action of AST VII and its analogs on their cellular activity and encourage further development of these compounds as a potential vaccine adjuvant.

## 2. Materials and Methods

### 2.1. Reagents, Cells, and Mice

AST VII was donated by Bionorm Natural Products, Izmir, Turkey. QS-21 was purchased from Desert King International (San Diego, CA, USA). GM-CSF (Granulocyte Macrophage Colony Stimulating Factor) (R&D Systems, Minneapolis, MN, USA), M-CSF (Macrophage Colony Stimulating Factor) (PeproTech, London, UK), LPS (Lipopolysaccharide) (invivogen, San Diego, CA, USA), Cytofix-fixation buffer (BD Bioscience, San Jose, CA, USA), Mouse IL-12p70, IL-2 ELISA kits (eBioscience), Human IL-2, IL-4, IL-17A, TNF-α, IFN-γ, and IL-1β ELISA kits (eBioscience, San Diego, CA, USA), and EasySep Mouse Naive CD4^+^ T cell and CD8^+^ T cell Isolation Kit (StemCell Technologies, Vancouver, BC, Canada) were purchased.

Mouse bone marrow-derived dendritic cells (BMDCs) were maintained in R5 medium containing RPMI 1640 medium with 5% Fetal Bovine Serum and 50 U/mL Penicillin/streptomycin (Gibco, Waltham, MA, USA) with 5 ng/mL GM-CSF. Mouse bone marrow-derived macrophages (BMDMs) were cultured in R5 medium + L929 medium (1:1) supplemented with 10 ng/mL M-CSF. 

C57BL/6 and BALB/c mice were housed in the animal facility of the Izmir Biomedicine and Genome Center (IBG). The mice were maintained in groups of 5 under standard conditions of temperature 22 ± 1 °C with regular 12 h light and 12 h dark cycles and free access to standard laboratory food and water. The experimental protocols were approved by the Local Ethics Review Committee for Animal Experimentation of IBG.

### 2.2. Semi-Synthesis of AST VII Derivatives and Their Physicochemical Characterization

The synthesis, purification, and structure elucidation of all newly synthesized compounds are described in detail in Appendix A. To determine critical micelle formation, 1 mg/mL dansyl chloride dissolved in acetone (20 μL) was transferred to 96-well plates and left to evaporate. A series of concentrations of AST VII and its derivatives (0.01 μM to 20 mM) was added into solvent-evaporated dansyl chloride and incubated overnight by shaking in the dark and at room temperature. The fluorescent intensity was measured using a microplate reader at excitation and emission wavelengths filter of 335 nm and 518 nm, respectively (Thermo, Varioskan Flash).

To assess the particle size distribution and zeta potential of the self-assembled structures, dynamic light scattering measurement was performed using a Zetasizer Nano ZS at a 173° scattering angle at 25 °C. AST VII (4000 μM), DC-AST VII (10 μM), and DAC-AST VII (10 μM) were dissolved in water, and the particle sizes were shown as the number of the particle distribution. The morphology of self-assembled structures was examined using a JEOL JEM 1220 electron microscope with an acceleration voltage of 100 kV. Samples were prepared by placing a drop of sample onto a formvar/carbon-coated copper grid and allowed to dry before imaging.

### 2.3. Cytotoxicity Evaluation of AST VII Analogs

Human endometrial carcinoma (HeLa, ATCC, CCL-2), human lung adenocarcinoma (A549, ATCC, CCL-185), human prostate carcinoma (Du145, ATCC, HTB-81), human ductal carcinoma (HCC-1937, ATCC, CRL-2336), human breast adenocarcinoma (MCF-7, ATCC, HTB-22), and human lung fibroblasts (MRC-5, ATCC, CCL-171) were obtained from the American Type Culture Collection and maintained as exponentially growing monolayers by culturing according to the supplier’s instructions. For the cytotoxicity analysis, each cell line was exposed to the compounds (DC-AST VII and DAC-AST VII) at a final concentration of 2, 4, 8, 16, or 32 μM for 48 h. After a 48 h incubation, the cell viability was analyzed using the MTT assay (Sigma Aldrich, St. Louis, MI, ABD) according to the manufacturer’s instructions. DMSO was used as a negative control. Absorbance was measured with a microplate reader at 570 nm (Varioscan, Thermo Fisher Scientific, Waltham, MA, US). IC50 values, defined as the compound concentration at which the response level was reduced to half its maximum, were calculated using GraphPad Prism 6 (San Diego, CA, USA).

### 2.4. Evaluation of Hemolytic Activities of AST VII Analogs

A hemolytic activity assay was performed with human red blood cells obtained from healthy volunteers [23]. A total of 7 mL of blood was washed and centrifuged three times at 2000× *g* for 5 min with sterile PBS. The cell suspension was diluted in saline solution with a final concentration of 0.5%. A total of 0.01 mL of the cell suspension was mixed separately with 0.01 mL saline solution of AST VII and derivatives at varying concentrations (0, 25, 50, or 250 µg/mL). The mixtures were incubated for 30 min at 37 °C and centrifuged at 800× *g* for 10 min. The free hemoglobin content of the supernatants was measured spectrophotometrically at 540 nm. Saline and distilled water were referred to as minimum and maximum hemolytic controls. 

### 2.5. Human Whole Blood (hWB) Stimulation Assay

The human whole blood (hWB) stimulation assay was performed to demonstrate the cytokine release profiles of AST VII analogs in parallel to AST VII and QS-21. Heparinized whole blood from healthy volunteers was supplemented in 1:10 and 1:20 with RPMI-1640 medium, 100 U/mL penicillin/streptomycin, and 10% fetal bovine serum. PMA (phorbol 12-myristate 13-acetate) (50 ng/mL) and ionomycin (400 ng/mL) (Sigma, St. Louis, MI, USA) stimulated hWB were treated with the following compounds: AST-VII, DC-AST VII, DAC-AST VII, and QS-21 at the concentrations of 2, 4, 8, 16, and 32 μg/mL. The cells were incubated at 37 °C in 5% CO_2_ for 48 h. Supernatants of each well were collected and stored at −20 °C for ELISA.

### 2.6. Generation of Bone Marrow-Derived Dendritic Cells (BMDCs) and Macrophages (BMDMs)

BMDCs were prepared as described previously [28]. Briefly, femurs and tibias of C57BL/6 and BALB/c mice were collected and flushed with sterile DPBS twice. The resulting bone marrow cells were resuspended in R5 medium (RPMI 1640 medium containing 5% heat-inactivated FBS, 2 mM L-glutamine, 50 U/mL penicillin/streptomycin) plus 5 ng/mL GM-CSF. A total of 2 × 10^6^ cells per bacteriological culture plate were cultured for 10 days. The cells were transferred to a new plate on day 6, and fresh medium was added to the culture on days 3 and 8. Non-adherent and loosely adherent cells were collected on day 10, and the cells were verified to be 80% CD11c^+^ using flow cytometry analysis. BMDMs were prepared as described previously [29]. Briefly, bone marrow cells from C57BL/6 mice were resuspended in R5 medium containing 10 ng/mL M-CSF and cultured for a day in a tissue culture plate. The next day, the non-adherent cells were collected and transferred to a 6-well low attachment plate in 4 mL R5 medium containing 30% L929 conditioned medium and 10 ng/mL M-CSF per well for 7 days. Fresh medium was added to the culture on day 6. After that, the cells were purified using centrifugation over Ficoll-Paque plus. The cells were verified to be >90% CD11b^+^ F4/80^+^ using flow cytometry analysis.

### 2.7. Stimulation of BMDCs and BMDMs with AST VII, DC-AST VII, and DAC-AST VII

BMDCs were suspended in R5 medium and cultured in 96-well plates at the concentration of 2.5 × 10^5^ BMDCs/well in 200 μL. AST VII and its derivatives were dissolved in DMSO. Different concentrations of AST VII (1 to 64 μM), DC-AST VII, and DAC-AST VII (2 to 20 µM) were added to the appropriate wells and incubated for 24 h at 37 °C in a 5% CO_2_ incubator. In the presence of LPS (10 ng/mL), BMDCs were treated with AST VII, DC-AST VII, and DAC-AST VII (0.5, 2, 2.5, 5, 10 μM) and incubated for 24 h. The cells were analyzed for the expression levels of MHC II, CD86, and CD80 using flow cytometry and the supernatant was analyzed for IL-12 production using ELISA. BMDMs were plated as 1 × 10^5^ BMDMs/well in 200 μL R5 medium. The cells were treated with LPS (10 ng/mL) plus different concentrations of AST VII, DC-AST VII, and DAC-AST VII (2, 5, 10 μM). Six hours later, the supernatant was collected and the IL-1β cytokine release was analyzed using ELISA.

### 2.8. Naive CD4+ and CD8+ T Cell Isolation

Spleens from C57BL/6 mice were disrupted in DPBS containing 2% FBS. Single splenocyte suspensions were obtained by passing the cells through a 70 μm mesh nylon strainer, followed by centrifugation at 300× *g* for 10 min three times. The cell pellet was resuspended in DPBS containing 2% FBS with a final concentration of 1 × 10^8^ nucleated cells/mL. Naive CD4^+^ and CD8^+^ T cells were isolated with negative selection using StemCell EasySep^TM^ Mouse Naive CD4^+^ and CD8^+^ T Cell Isolation Kit following the manufacturer’s instructions. 

### 2.9. Mixed Lymphocyte Reaction (MLR)

BMDCs generated from BALB/c mice were plated in 96-well plates as 2 × 10^4^ cells/well in 100 μL R5 medium. The cells were treated with LPS (10 ng/mL), AST VII (5 μM), DC-AST VII (5 and 10 μM), DAC-AST VII (5 and 10 μM) and incubated for 24 h at 37 °C in a 5% CO_2_ incubator. The next day, followed by washing with DPBS, naive CD4^+^ and CD8^+^ T cells isolated from C57BL/6 were added to the 96-well plates in a 1:5 ratio (BMDCs: T cells). The cells were incubated for 3 days at 37 °C in a 5% CO_2_ incubator. CD4^+^ and CD8^+^ T cell activation was assessed by the expression of CD44 (anti-CD44-APC, IM7, 103011, Biolegend, San Diego, CA, USA) using flow cytometry. The cell supernatant was collected and stored at −20 °C for subsequent cytokine analysis.

### 2.10. Flow Cytometry

For the staining of cell surface molecules, cells were resuspended in FACS buffer (PBS, 1% BSA, 0.025% NaN_3_). A Zombie UV Fixable Viability Kit (BioLegend, San Diego, CA, USA) was used to determine cell viability for flow cytometry experiments. The viability of the BMDCs treated with AST VII and derivatives relative to control was over 88% for all experiments. 

Fcγ receptors were blocked with 1/200 diluted anti-mouse CD16/32 antibodies (Fc block) (Tonbo Biosciences, San Diego, CA, USA) for 10 min on ice. BMDCs and splenocyte single-cell suspensions were stained with the following: 1/400 diluted fluorochrome-conjugated anti-mouse Abs: anti-CD11c-APC (N418, 117309), anti-MHC (I-A/I-E)-PE-Cy7 (M5/114.15.2, 107629), anti-MHC (I-A/I-E)-APC-Fire750 (M5/114.15.2, 107651), anti-CD80-Pacific Blue (16-10A1, 104723), anti-CD86-PerCp Cy5.5 (GL-1, 105027), anti-CD86-PE-Cy7 (GL-1, 105013), anti-F4/80-PE (BM8, 123109), and anti-CD11b-PerCp-Cy5.5 (M1/70, 101227) (Biolegend, San Diego, CA, USA) and incubated in the dark on ice for 40 min. The cells were fixed with Cytofix-fixation buffer (BD Biosciences) for 15 min on ice and cell fluorescence was assessed using Fortessa (BD Biosciences). The data were analyzed with FlowJo software version 10. The gating strategy for BMDCs was FSC-A: SSC-A > singlets > CD11c^+^ MHC II^+^ > MHC II^+^ CD86^+^, MHC II^+^ CD80^+^. The gating strategy for DCs from splenocytes was FSC-A: SSC-A > singlets > F4/80^−^ > CD11b^+^ CD11c^+^ > MHC II^+^ CD86^+^ > MHC II^+^ CD80^+^. The gating strategy for lymphocytes was FSC-A: SSC-A > singlets > live-dead > CD4^+^ CD8^−^; CD4^−^CD8^+^ > CD44^+^.

### 2.11. ELISA

The cell supernatants were collected, and the concentration of human IL-2, IFN-γ, IL-4, IL-17A, TNF-α, and IL-1β and of mouse IL-12 and IL-1β were assessed using ELISA (eBioscience, San Diego, CA, USA) in accordance with the manufacturer’s instructions.

### 2.12. Statistical Analysis

Data for all experiments were analyzed with Prism software (GraphPad). The data are expressed as mean ± SD. Unpaired Student’s *t*-test, one-way ANOVA, and Dunnett’s or Tukey’s multiple comparisons tests were used for the comparison of experimental groups when appropriate. *p*-values of less than 0.05 were considered statistically significant. 

A statistical approach of principal component analysis (PCA) was used to interpret the relationship between cytokine production and the compound treatments. PCA was performed using Prism software (GraphPad). 

## 3. Results and Discussion

### 3.1. Semi-Synthesis of Astragaloside VII (AST VII) Derivatives

There is no rational path for designing new saponin adjuvants originating from *Astragalus* saponins to improve their immunomodulatory activities. Therefore, we made a preliminary attempt to semi-synthesize AST VII, a promising adjuvant candidate, to obtain diverse AST VII variants and evaluate their immunomodulatory activities. We carried out the synthetic strategy illustrated in Figure 1.

AST VII contains the primary alcohols on glucose moieties extending from the aglycone backbone. To oxidize these primary alcohols to carboxylic acid, the TEMPO-mediated oxidation reaction [30] was followed and at the end, dicarboxylic AST VII (DC-AST VII) was synthesized at 57% yield. In a further step, the glucuronic acid carboxyl of DC-AST VII was conjugated with free dodecylamines via amide bond formation to derive dodecylamine-conjugated AST VII (DAC-AST VII). This reaction was carried out with EDC/HOBt coupling reagents at 28% yield. Semi-synthetic compounds were extracted and further purified using column chromatography. The chemical structures of newly synthesized derivatives were elucidated using 1D and 2D NMR [Varian MERCURY plus AS400 (400 MHz)] and TOF-MS spectroscopy (Agilent 1200/6530) (Appendix A). DC-AST VII, having two glucuronic acids at C6 and C25 on the aglycone structure, represents a more hydrophilic structure than AST VII. On the other hand, DAC-AST VII with the dodecylamide hydrocarbon chains extending from the glucuronic acid moieties at C6 and C25 on the aglycone forms the hydrophobic analog of AST VII.

### 3.2. Cytotoxic and Hemolytic Activities of AST VII Analogs on Human Cancer Cell Lines and Erythrocytes

The cytotoxic and hemolytic activities of the newly synthesized AST VII analogs were investigated. The lead compound, AST VII, did not reveal any cytotoxic or hemolytic activities in previous studies [23]. Five cancer cell lines (A549, Du145, HCC-1937, HeLa, MCF-7) and one control cell line (MRC-5) were used to screen the cytotoxic properties of DC-AST VII and DAC-AST VII using an MTT assay. There was no/slight reduction in the cell viability following DC-AST VII treatment compared to DMSO (vehicle control) in cancerous cell lines. Moreover, DC-AST VII at 2 µM concentration statistically enhanced cell proliferation in MRC-5 cells compared to DMSO. In contrast to DC-AST VII, DAC-AST VII demonstrated cytotoxicity in all cell lines in a dose-dependent manner with the following IC50 values: 29.51 µM for A549, 7.91 µM for Du145, 12.06 µM for HCC-1937, 11.6 µM for HeLa, 9.79 µM for MCF-7, and 17.04 µM for MRC-5 (Appendix A). While DC-AST VII did not demonstrate cytotoxicity in cancerous cell lines as AST VII, DAC-AST VII showed cytotoxicity, indicating that lipophilicity is an important parameter for its cytotoxicity. The lysis of erythrocytes is one of the biological characteristics of saponins, and it is a limiting factor for the utilization of these compounds in clinics [31]. Next, the hemolytic activity of AST VII analogs on human erythrocytes was investigated. Serially diluted (1/5) concentrations (250 µM to 0.4 µM) of DC-AST VII and DAC-AST VII were analyzed. No hemolysis was observed in human erythrocytes following treatment with DC-AST VII and DAC-AST VII (Appendix A). Overall, AST VII and its derivatives have an advantage over other saponin compounds as hemolytic activity is one of the drawbacks in vaccine development. 

### 3.3. Evaluation of Immunomodulatory Activities of AST VII and Its Derivatives Based on Cytokine Release on Human Whole Blood

The human whole blood (hWB) stimulation assay is a simple and effective approach to evaluate the immunomodulatory activities of the test compounds as it contains diverse immune cells, such as T cells, B cells, NK cells, monocytes, and granulocytes [32]. To address what type of cellular immune responses (Th1/Th2/Th17) was induced by AST VII and its analogs, the cytokine release profiles (IL-2, IFN-γ, IL-17A, IL-1β, TNF-α, IL-4) of PMA-ionomycin (P+I)-stimulated hWB were investigated. 

As PMA (protein kinase C activator) is commonly used with ionomycin (calcium ionophore) to activate T cells by inducing NF-κB and NFAT transcription factors, and successively leading to the production of cytokines [33], hWB was diluted in a 1/10 ratio with RPMI-1640 medium and stimulated with PMA (50 ng/mL) and ionomycin (400 ng/mL) in the presence or absence of AST VII or its derivatives. QS-21, a widely used saponin based adjuvant was also tested for comparison. Surprisingly, in the absence of the saponin compounds, we did not observe any significant production of IL-1β and IL-17A cytokines compared to the control group (untreated cells) (Appendix A). This was thought to be caused by a too high cell concentration in the 1/10 diluted hWB. Therefore, for later analysis, we decided to dilute hWB in a 1/20 ratio. When the results are analyzed in detail, AST VII (up to 2.24 fold), DC-AST VII (2.52 fold), and DAC-AST VII (3.32 fold) substantially increased IL-1β production. Overall, the increase in IL-1β production induced by AST-VII and its derivatives was higher compared to QS-21 (Appendix A). In addition, IL-1β helps naive T cells to differentiate into Th17, promotes IL-17-producing memory CD4^+^ T cells, and is a profound inducer of IL-17^+^ T cell differentiation along with TGF-β [34,35]. IL-17A levels were induced by AST VII (up to 5.05 fold) and QS-21 treatments compared to P+I treatment alone (Appendix A). Nalbantsoy et al. reported that AST VII in the presence of LPS was able to stimulate TGF-β production [24]. The need for TGF-β may be caused by a difference in IL-17A production between AST VII and its derivatives.

In contrast to the 1/10 diluted hWB, the 1/20 diluted hWB demonstrated statistically significant cytokine production after P+I treatment compared to the control group (untreated cells). AST VII and DC-AST VII did not enhance IL-1β production, whereas DAC-AST VII (4.2-fold) remarkably boosted IL-1β levels in a dose-dependent manner compared to P+I alone (Figure 2A). The production of TNF-α, IFN-γ, and IL-2 cytokines was suppressed following treatment with AST VII, DC-AST VII, or DAC-AST VII compared to P+I (Figure 2B–D). This result demonstrated contradicting data with our previous studies [23,24,36] showing that AST VII induced the production of Th1 cytokines (IL-2 and IFN-γ). Activation of T cells by PMA/ionomycin successively leads to the great production of cytokines [33]. In addition to PMA/ionomycin stimulation, T cells can also be activated via an antigen-dependent pathway, which involves a CD3/T-cell receptor complex together with signals provided by costimulatory molecules such as CD28 [37]. The effect of AST VII and its derivatives on TCR-mediated T cell activation will be investigated in future studies. 

In terms of Th2 type cytokines, AST VII and its derivatives did not stimulate IL-4 production, in parallel with previous reports [23,24], although some triterpenoid saponins, such as Ginsenoside Rg1 and Ginsenoside Rd, were shown to induce Th2 cytokines. As the indicator of the Th17 response, IL-17A was not detected in the cell culture supernatant following AST VII treatment in 1/20 diluted hWB, possibly resulting from the consumption of IL-17A via IL-17 receptor. This consumption by a specific receptor in vitro has also been suggested as the possible reason for undetectable IL-4 in cell culture supernatant [38,39].

Variability in cytokine type and levels obtained by treating saponin compounds showed complex and multifaceted interactions that were difficult to interpret. Therefore, principal component analysis (PCA), a multivariate data analysis technique, was employed to see the correlations in the cytokine response. The PCA biplot for 1/10 diluted hWB demonstrated the clusters of AST VII—QS-21 and DC-AST VII—DAC-AST VII (Appendix A), indicating similar responses were obtained in the clustering groups. Moreover, the IL-1β response was inversely correlated to IL-17A as the angle between their vector projections was close to 180°. In the PCA biplot for the 1/20 diluted hWB, IL-1β, TNF-α, and IFN-γ provided positive loadings for principal component (PC) 1 (correlation coefficient r > 0.5 (Appendix A)) (Figure 2E). 

IL-1β, TNF-α, and IFN-γ are pro-inflammatory cytokines and have a role in potentiating an immune response by improving antigen presentation and activating related immune cells [40]. There was also a positive correlation between these cytokines in the PCA biplot. IL-2 is a pleiotropic cytokine and can function as a master regulator that modulates many cytokines to influence the cell priming for differentiation and maintains a differentiated state [41]. As IL-2 and IL-1β have an inverse loading on PC2, we can conclude that these cytokines have a role in modulating the immune response produced by AST VII and DAC-AST VII. Overall, these data show that when co-treated with PMA-ionomycin, AST VII and its derivatives promoted the pro-inflammatory cytokine production of hWBs.

### 3.4. IL-1β Secretion Following Treatment with AST VII and Its Derivatives in BMDCs and BMDMs

The prominent induction of IL-1β leads us to investigate the role of these compounds in innate cell activation, such as dendritic cells and macrophages [42]. Bone marrow cells from C57BL/6 mice were differentiated into bone marrow-derived dendritic cells (BMDCs) or bone marrow-derived macrophages (BMDMs) in vitro with GM-CSF or M-CSF, respectively. IL-1β production and its secretion from these cells progress in three steps: (a) priming: production of pro-IL-1β; (b) processing of pro-IL-1β to mature IL-1β; and (c) release of IL-1β from the cells [43]. As the priming step requires stimulation of the TLR/NF-κB pathway, and AST VII alone cannot activate NF-κB [44], LPS was used to activate pro-IL-1β in BMDCs and BMDMs [45]. Upon co-treatment with LPS (10 ng/mL), AST VII and its derivatives significantly boosted IL-1β secretion in BMDCs compared to LPS alone, which has been reported to be a weak stimulus for IL-1β secretion in dendritic cells [46]. When the comparison was made between AST VII and its derivatives, AST VII (10 μM) substantially augmented IL-1β production compared to DC-AST VII (10 μM) and DAC-AST VII (10 μM) (Figure 3A). In BMDMs, AST VII and DC-AST VII demonstrated a statistically significant increase in IL-1β compared to LPS alone, whereas IL-1β production after DAC-AST VII treatment was below the detection limit (Figure 3B). These data indicate that all compounds for BMDCs, AST VII, and DC-AST VII for BMDMs boosted the production of IL-1β in the presence of LPS. Moreover, these compounds were more effective in dendritic cell activation compared to macrophages as the level of IL-1β produced in dendritic cells was much higher than in macrophages.

When the effect of *Astragalus* saponins on the production of IL-1β was searched, Astragaloside IV (AST IV), another cycloartane type triterpene glycoside that contains xylose and glucose moieties at C-3 and C-6 on the aglycone backbone, inhibited the release of IL-1β, and the activation of NF-κB in LPS induced epithelial cells in vitro and in vivo. Moreover, treatment with AST IV alone did not alter the gene expressions of pro-inflammatory cytokines and adhesion molecules [47,48]. According to these reports, the presence of glucose at C-25 on the sapogenin backbone of AST VII possibly changes the ability to induce IL-1β production, signifying the importance of the tridesmosidic nature of AST VII.

On the other hand, QS-21, a *Quillaja* saponin, in the presence of the TLR4 agonist MPLA was reported to enhance caspase-1/11 and NLPR3-dependent IL-1β production. Moreover, Marty-Roix et al. investigated IL-1β production in macrophages following treatment with different types of saponins such as QS-21, digoxin, sapindoside A, hedaracoside C, or β-escin in combination with MPLA. Only QS-21 elicited an IL-1β response, indicating that the IL-1β response and inflammasome activation could be specific to *Quillaja* saponins. To test this hypothesis, *Quillaja* saponin (Quil-A) and VET-SAP^®^ were administrated to BMDCs/BMDMs, and both showed NLRP-3-dependent IL-1β secretion [12]. In addition to *Quillaja* saponins, AST VII and its derivatives also boosted the production of IL-1β, and inflammasome activation may have a role in the underlying production of IL-1β with the treatment of AST VII and derivatives. 

### 3.5. AST VII and Its Derivatives Induced Dendritic Cell Maturation and Activation

IL-1β can act as a maturation factor for DCs in terms of IL-12 production and expression of CD86, MHC I, and ICAM-1 [49]. Given the enhanced production of IL-1β in hWB, BMDCs, and BMDMs by AST VII and its derivatives, we next investigated if these compounds affect dendritic cell maturation and activation. The maturation state of BMDCs was evaluated with the analysis of MHC II, CD86, and CD80 expression using flow cytometry and the secretion of IL-12 using ELISA. 

Firstly, treatment with AST VII alone induced neither the expression of MHC II, CD86, or CD80 nor the production of IL-12 in BMDCs even at high concentrations (up to 60 μg/mL). To investigate whether AST VII enhances dendritic cell maturation/activation with the help of other cell types or secreted cytokines in the microenvironment, a single-cell suspension of splenocytes from C57BL/6 mice was treated with AST VII alone at concentrations of 5, 10, or 20 μg/mL. However, AST VII stimulation did not result in the upregulation of MHC II, CD80, or CD86 (Appendix A). These findings demonstrate that AST VII alone does not affect the maturation/activation of dendritic cells in vitro.

As robust IL-1β production was obtained by co-treatment with LPS, BMDCs were incubated with LPS (10 ng/mL) with or without AST VII at the concentrations of 2, 5, or 10 μM for 24 h. Compared to LPS alone, AST VII plus LPS significantly increased the expression on BMDCs of MHC II (10 μM, Figure 4A), CD86 (all doses, Figure 4B), and CD80 (2 μM, 10 μM, Figure 4C). AST VII at 10 μM was more effective than 2 μM or 5 μM in the upregulation of both CD86 and CD80 (*p* < 0.05 and *p* < 0.01, respectively) (Figure 4B,C). In addition, compared to LPS alone, AST VII (10 μM) increased the frequencies of CD86^+^ and CD80^+^ BMDCs, whereas it did not alter the frequency of the MHCII^+^ population (Appendix A). 

In addition to the upregulation of the costimulatory molecules, the secretion of IL-12 is also one of the indicators of DC activation [50]. The production of IL-12 by BMDCs was significantly increased following co-stimulation with LPS and AST VII compared to LPS alone (Figure 4D). These data show that AST VII induces dendritic cell maturation and activation in cooperation with LPS (Figure 4E) and can act as the co-adjuvant. A previous study carried out by Marty-Roix et al. showed that QS-21 alone did not lead to the activation of murine BMDCs and BMDMs directly [12]. QS-21 in liposomal formulation with MPL, another TLR-4 agonist, augmented MHC II and CD86 expression on human monocyte-derived dendritic cells [11]. Therefore, our study shows that the AST VII influence on dendritic cell activation is comparable with QS-21.

To evaluate the impact of DC-AST VII and DAC-AST VII on dendritic cell maturation/activation, BMDCs were treated with the compounds for 24 h in the absence/presence of LPS. Without LPS stimulation, DC-AST VII at the concentration of 5 and 10 μM increased the expression of MHC II, CD86, and CD80 on BMDMs compared to the control group (untreated) (Figure 5A–C). Moreover, in the absence of LPS, DC-AST VII (5 and 10 μM) increased the frequencies of CD86^+^ and CD80^+^ BMDCs, whereas it did not alter the frequency of the MHCII^+^ population (Appendix A). In general, DAC-AST VII treatment alone did not demonstrate an increase in the expression of MHC II, CD86, and CD80 (n.s.). Interestingly, MHC II expression on BMDCs was decreased by DAC-AST VII at the concentration of 10 μM compared to the control group (Figure 5A). Similarly, DAC-AST VII treatment did not increase the frequencies of MHC II^+^, CD86^+^, and CD80^+^ BMDCs (Appendix A).

Next, to determine whether the stimulation of DC-AST VII and DAC-AST VII with LPS would alter their adjuvant activity on BMDCs, the cells were co-treated with DC-AST VII or DAC-AST VII at 10 µM along with LPS (10 ng/mL). In the presence of LPS, DC-AST VII and DAC-AST VII reduced the expression of MHC II, CD86, and CD80 compared to LPS alone (Figure 5D–F). The reduction by DAC-AST VII was more prominent compared to DC-AST VII. Furthermore, LPS and DC-AST VII or DAC-AST co-treatment did not alter the frequency of MHC II^+^ BMDCs compared to LPS alone (Appendix A).

However, DC-AST VII and DAC-AST VII reduced the frequencies of CD86^+^ and CD80^+^ BMDCs (Appendix A). Overall, DC-AST VII (10 μM) alone activated dendritic cells, but the effect of DC-AST VII was diminished in the presence of LPS. The calculated partition coefficients (clogP) clearly demonstrate a hydrophilic order of test compounds (DC-AST VII (clogP: 0.34) >AST VII (clogP: 1.29) >DAC-AST VII (clogP: 11.41)), which implies that the polarity of the compounds is indeed an important factor for dendritic cell activation.

A contrary result was obtained with the co-treatment of AST VII and its derivatives with LPS. Thus, we investigated whether the reduction in dendritic cell activation by co-treatment of AST VII derivatives and LPS was correlated with their self-assembled structure formation properties. As saponins are amphiphilic compounds, due to the presence of a lipid-soluble aglycone and water-soluble sugar chain [51], they have the tendency to form aggregates or micelles in aqueous solutions. Micelle formation is a concentration-dependent process that is characterized by a sharp transition at the critical micelle concentration (CMC). Below the CMC, the compounds are unassociated monomers or form a monolayer at the air-solvent interface and some of them begin to form aggregates (C > CAC (Critical aggregation concentration)), whereas above the CMC, the compounds assemble to form micelles (Figure 6A) [52].

To determine the self-assembled structure formation properties of *Astragalus* saponins, firstly, the compounds at various concentrations were mixed with fluorescence dye (dansyl chloride) overnight and the relative fluorescence intensity at 508 nm was measured. Fluorescence intensity was plotted as a function of the logarithm of the concentrations of compounds, and each CMC value was calculated using the intersection of two linear lines. The CMC values for AST VII, DC-AST VII, and DAC-AST VII were calculated as 3.37 mM, 7.3 μM, and 0.047 μM, respectively (Figure 6B).

Based on the CMC values, it can be concluded that AST VII is mainly a monomer form in the cell culture, whereas DC-AST VII and DAC-AST VII form self-assembled structures at the treatment concentrations. On the other hand, LPS has a very low critical micelle concentration of 10^−9^ mol [53]. Only its monomers can bind to TLR4/MD2 heterodimer, and further transduce the signal [54]. Thus, AST VII may be facilitating the transfer and binding of LPS to TLR4/MD2. In the case of AST VII derivatives, micelle/aggregate formation can lead to the encapsulation of LPS as the hydrophobic properties of LPS enables itself to be entrapped into self-assembled structures easily. The difficulties in the release of LPS from these structures may lead to the reduction of dendritic cell activation. As self-assembled structures derived from DAC-AST VII contain dodecyl acyl chains in their inner core, the stability of LPS entrapped in the structures will be increased, causing a stronger reduction of the dendritic cell activation compared to DC-AST VII.

The particle size distribution of these compounds at the concentration above their CMC values was measured using dynamic light scattering (DLS). The hydrodynamic particle size for self-assembled structures produced by AST VII (4000 μM), DC-AST VII (10 μM), and DAC-AST VII (10 μM) in aqueous solution was analyzed as 95 nm, 74 nm, and 30 nm (Figure 6C), respectively, and demonstrated a polydispersity index below 0.5, which was attributed to a homogeneous size distribution. The morphology of the individual self-assembled structures existing dominantly on the TEM grid is given in Figure 6D. The clusters of small micellar structures or aggregates on the TEM grid can lead to the formation of larger particles than the particles analyzed with DLS. These data show that AST VII and its derivatives formed self-assembled structures in an aqueous solution above their CMC values, and the immunomodulatory activity of these compounds was altered depending on their micelle/aggregate formation concentration.

### 3.6. AST VII and Its Derivatives Activated CD4^+^ and CD8^+^ T Cells in Mixed Leukocyte Reaction (MLR)

To determine the impact of dendritic cell activation by AST VII and its derivatives on the ability of T cell priming, we performed a mixed leukocyte reaction (MLR) assay. MLR is a simple and effective in vitro model to investigate T cell activation and proliferation [55]. BMDCs generated from the bone marrow cells of BALB/c mice were treated with LPS, AST VII, DC-AST VII, or DAC-AST VII for 24 h. The next day, naive CD4^+^ or CD8^+^ T cells isolated from the spleens of C57BL/6 mice were co-cultured with BMDCs for 3 days.

The activation state of T cells was evaluated by measuring CD44 expression on T cells using flow cytometry as CD44 is expressed on activated T cells. In addition, CD44 enhances the stability of the DC-T cell interaction and T cell proliferation to TCR signaling in vitro [56,57].

All test compounds increased the frequency of CD44^+^, CD8^+^, and CD4^+^ T cells (Appendix A). However, some of them demonstrated a statistical increase in the expression of CD44 on CD4^+^ and CD8^+^ T cells based on their structural features (Figure 7A,B). In CD8^+^ T cells, LPS, LPS+AST VII (5 μM), and DAC-AST VII (10 μM) increased the expression of CD44 compared to the control (untreated) group. DAC-AST VII was more potent in the activation of CD8^+^ T cells than LPS alone. Interestingly, AST VII alone and DC-AST VII did not significantly augment CD44 on CD8^+^ T cells. Moreover, there were no statistically significant differences in CD4^+^ T cell activation between either LPS alone and LPS+AST VII or AST VII alone and LPS+AST VII (n.s.) (Figure 7A).

In the case of CD4^+^ T cells, although LPS alone or AST VII alone did not have any statistically significant effect, the AST VII and LPS co-treatment, DC-AST VII (10 μM), and DAC-AST VII (10 μM) augmented CD44 expression compared to the control (untreated) group (Figure 7B). Moreover, DC-AST VII (10 μM) was more effective in CD4^+^ T activation compared to DAC-AST VII (10 μM).

Our data demonstrate that the BMDCs treated with LPS+AST VII and DAC-AST VII activated both CD4^+^ and CD8^+^ T cells. Imine-formation between the carbonyl group on DC-AST VII and DAC-AST VII and amino groups could have a role in the activation of T cells. This imine or Schiff-base formation by reacting carbonyl groups with amino groups, likely from the CD2 receptor on T cells, delivers a signal that replaces the one derived from the interaction between CD80/86 ligands and the CD28 receptor on dendritic cells and T cells, respectively [58]. Taken together, these data indicate that AST VII needs LPS to activate T cells, but AST VII analogs can activate T cells without LPS co-treatment, suggesting different action mechanisms for AST VII and its derivatives. 

DAC-AST VII, the lipophilic analog of AST VII because of dodecylamine conjugation, was better at CD8^+^ T cell activation, while DC-AST VII was more potent in CD4^+^ T cell activation. It was reported that dodecylamine-conjugated QS-21, namely GPI-0100, induced Th1 and CTL responses. The lipophilic character of GPI-0100 was proposed to enable the compound to open a transient pore for the delivery of exogenous proteins to the cytosol and further enhance MHC I-restricted CTL responses [16]. In another study, the QS-21 derivative 3, prepared by modification of the glucuronic acid carboxyl with ethylamine, was shown to enhance CTL responses comparable to QS-21 in mitomycin-C treated OVA-specific E.G7-OVA cells [59]. On the other hand, the QS-21 derivatives 2 and 4, prepared by the modification of glucuronic acid carboxyl with glycine and ethylenediamine, was shown to increase CTL response but not as high as QS-21. 

Increases in the lipophilicity of the compound influenced the CTL responses [59]. In this study, in a similar manner, we introduced dodecylamine onto glucuronic acid residues and increased the lipophilicity of AST VII and demonstrated that the CTL (CD8^+^ T cell) response was improved. This may be due to the enhanced translocation of the antigen into the cytosol, leading to more efficient antigen presentation by MHC I molecules to CD8^+^ T cells, which in turn results in TCR:MHC I engagement and upregulation of CD44. As CD44 expression can be regulated by pro-inflammatory cytokines such as IL-1β [60], the IL-1β secretion from dendritic cells might be the cause for the CD44 upregulation on T cells. Therefore, further studies are warranted to show the effects of IL-1β on CD44 gene regulation.

## 4. Conclusions

Adjuvants are one of the essential components in modern vaccines that are capable of initiating an innate immune response and subsequently an adaptive immune response [61]. The adjuvant potential of plant-derived compounds is widely investigated because of their abilities to enhance proper immune responses with limited toxicity [62]. In the present study, the immunomodulatory activities of the purified *Astragalus* saponin, Astragaloside VII (AST VII), and its newly synthesized analogs DC-AST VII and DAC-AST VII were investigated. Our study demonstrates that cycloartane-type saponins, AST VII and the newly synthesized derivatives DC-AST VII and DAC-AST VII, induce IL-1β production in hWB cells and BMDCs. Robust IL-1β production by these compounds was associated with the maturation and activation of dendritic cells, CD4^+^ and CD8^+^ T cells. The modification of the compounds toward more or less polar derivatives altered the activation status of CD4^+^ and CD8^+^ T cell responses. In addition to the potency to enhance immune responses, another critical factor for adjuvant development is the availability and ease of production of the adjuvant itself. Many saponin-based adjuvants suffer from low isolation yield from the plant material, inconsistent composition, and insufficient purity [63]. In contrast, AST VII can be isolated with large yields, is highly stable, highly soluble in water, has a slight hemolytic activity at high concentrations, and can readily be lyophilized. This makes AST VII a good alternative to other saponin-based adjuvants. This study is a first step to evaluate the mechanism of action of AST VII and subsequent studies toward a rational design of AST VII-based adjuvant analogs/vaccine formulations to be utilized in prophylactic and therapeutic vaccines. 

## 5. Patents

TR 2019 10003 A2 A METHOD FOR OBTAINING SAPONIN MOLECULES AND UTILIZING ACTIVE MOLECULES AS IMMUNOMODULATOR Application number: a 2019/10003. Publication: 22 January 2021.

## Figures and Tables

**Figure 1 vaccines-11-00495-f001:**
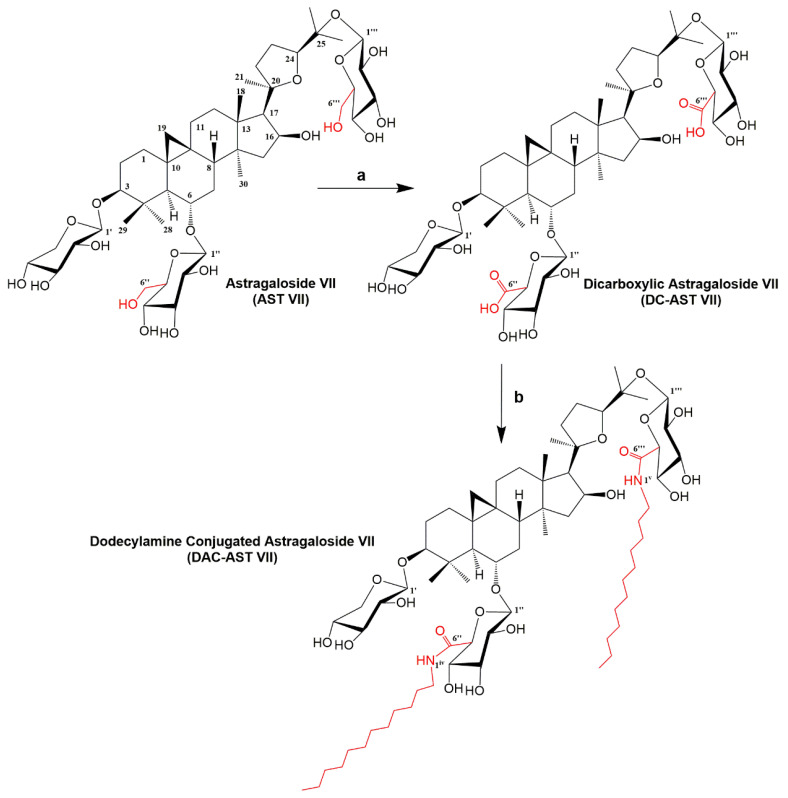
Semi-synthesis of AST VII derivatives. Reagents and conditions: (**a**) TEMPO, NaBr, NaOCl, H_2_O, 0 °C (57%) and (**b**) EDC/HOBt, DIPEA, Pyridine, 60 °C (28%).

**Figure 2 vaccines-11-00495-f002:**
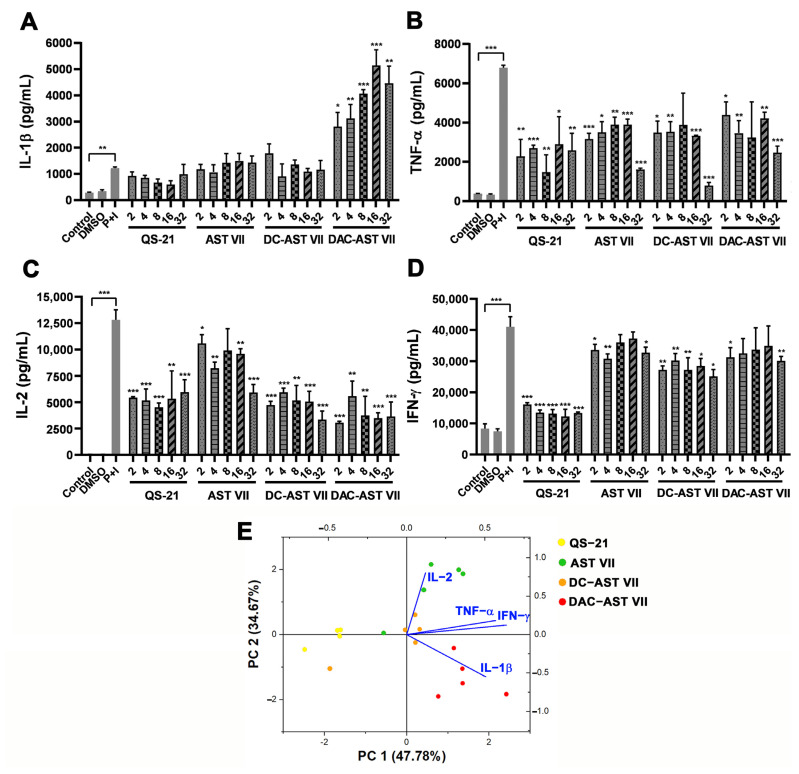
AST VII and its derivatives (DC-AST VII and DAC-AST VII) alter the production of pro-inflammatory and Th- mediated cytokines in hWB cells. Diluted hWB was co-treated with PMA (50 ng/mL)/ionomycin (400 ng/mL) and the following compounds: QS-21, AST VII, DC-AST VII, DAC-AST VII at the concentration of 2–32 µg/mL for 48 h. The supernatants were collected for the detection of cytokines using ELISA. (**A**) IL-1β, (**B**) TNF-α, (**C**) IL-2, and (**D**) IFN-γ in 1/20 diluted hWB. (**E**) Principal component analysis (PCA) biplot illustrating the cytokine responses and corresponding compound treatments in hWB. Data are projected onto the plane of the first two principal components (PCs) and colored by the different compounds. DMSO was used as vehicle control. Data shown are mean ± SD of triplicate determinations and representative of two independent experiments with similar results. Statistical analyses were performed between the control and P+I (PMA+Ionomycin) using Student’s t-test and P+I and treated groups using One-way ANOVA and Tukey’s multiple comparison tests. * *p* < 0.05, ** *p* < 0.01, *** *p* < 0.001, not statistically significant (ns).

**Figure 3 vaccines-11-00495-f003:**
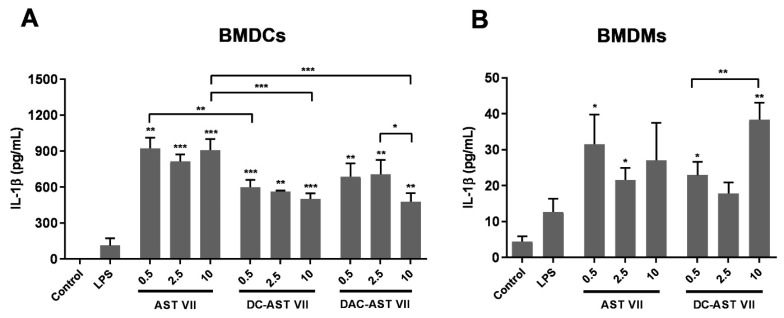
AST VII and its derivatives induce IL-1β secretion in BMDCs and BMDMs. (**A**) BMDCs and (**B**) BMDMs generated from bone marrow cells of C57BL/6 mice were treated with LPS (10 ng/mL) alone or LPS with AST VII, DC-AST VII, or DAC-AST VII at the concentrations of 0.5, 2.5, or 10 µM for 6 h. Unstimulated cells were used as the control. The cell culture supernatants were collected to analyze IL-1β concentrations using ELISA. Data shown are the mean ± SD of triplicates and representative of two independent experiments with similar results. Statistically significant differences in the intragroup were analyzed compared to LPS. Intragroup and intergroup comparisons were performed with one-way ANOVA and Tukey’s multiple comparisons test. * *p* < 0.05, ** *p* < 0.01, *** *p* < 0.001.

**Figure 4 vaccines-11-00495-f004:**
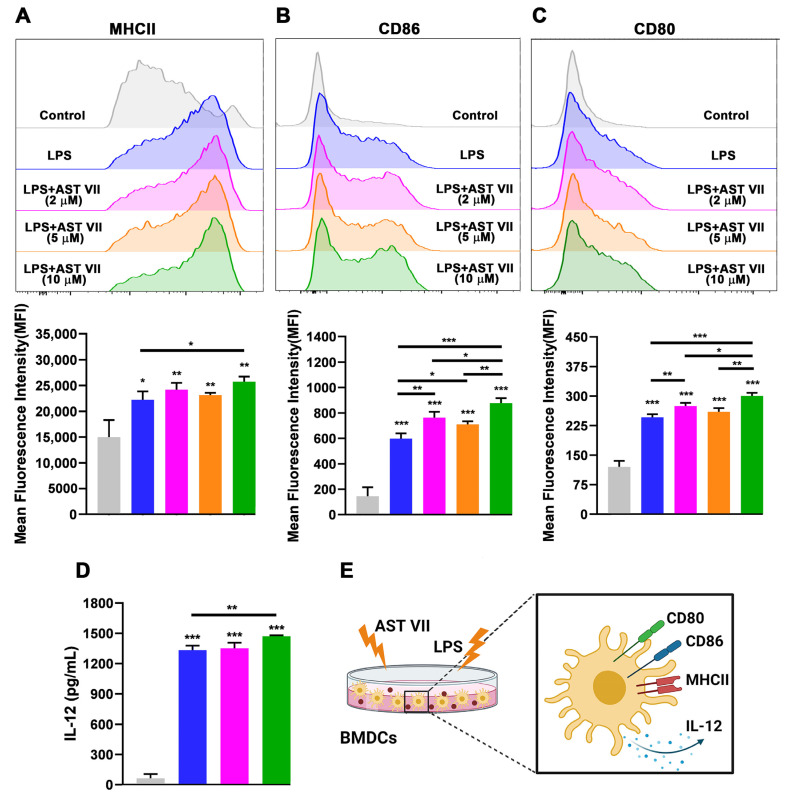
AST VII induces the maturation and activation of BMDCs in the presence of LPS. BMDCs generated from C57BL/6 mice bone marrow cells were co-treated with LPS (10 ng/mL) and AST VII at the concentrations of 2, 5, or 10 µM for 24 h. The expression of (**A**) MHCII, (**B**) CD86, and (**C**) CD80 on CD11c^+^MHCII^+^ BMDCs was determined using flow cytometry. For (**A**–**C**), representative histogram (top panel) and summary graphs (bottom panel) are shown. (**D**) IL-12 titers were measured using ELISA. (**E**) Schematic illustration of DC maturation/activation after the treatment with AST VII and LPS. Data shown are mean ± SD of triplicates and representative of three independent experiments with similar results. Statistically significant differences in the treated group were analyzed compared to the control (untreated cells). Intragroup and intergroup comparisons were performed using one-way ANOVA and Tukey’s multiple comparison test. * *p* < 0.05, ** *p* ≤ 0.01, *** *p* ≤ 0.001. Created with BioRender.com.

**Figure 5 vaccines-11-00495-f005:**
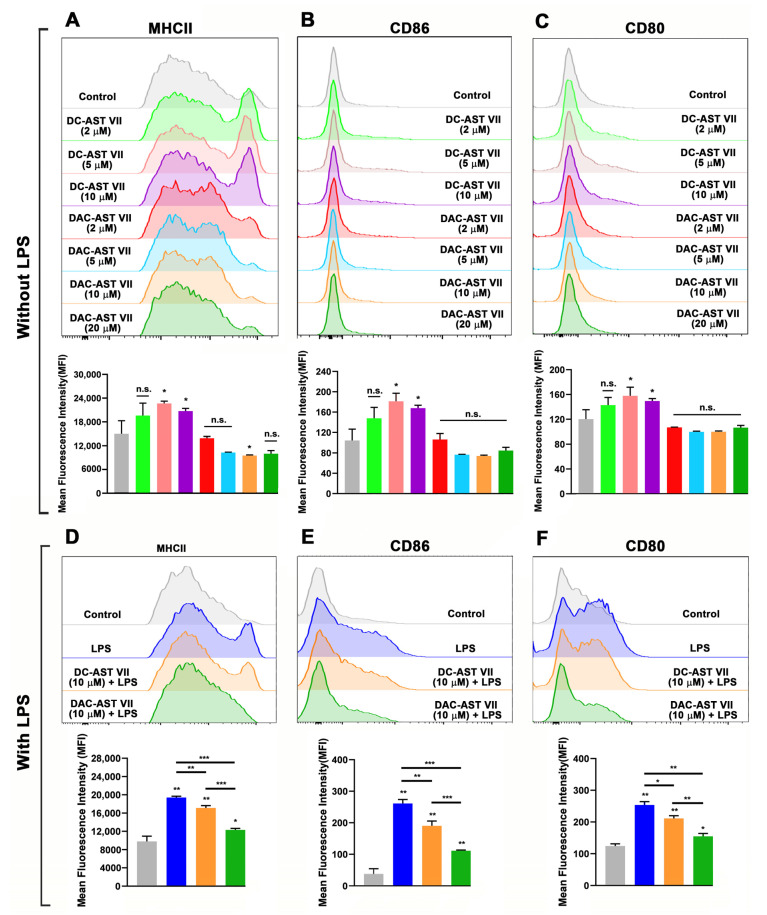
The effects of AST VII derivatives (DC-AST VII and DAC-AST VII) on the maturation and activation of BMDCs in the absence/presence of LPS. BMDCs generated from C57BL/6 mice bone marrow cells were treated with DC-AST VII or DAC-AST VII at the concentrations of 2, 5, 10, or 20 µM for 24 h. The expression of (**A**) MHCII, (**B**) CD86, and (**C**) CD80 on CD11c^+^MHCII^+^ BMDCs without LPS treatment and (**D**) MHCII, (**E**) CD86, and (**F**) CD80 on CD11c^+^MHCII^+^ BMDCs with LPS treatment was determined using flow cytometry. Representative histograms (top panel) and summary graphs (bottom panel) are shown. Data are shown as the mean and SD of triplicates and are representative of two independent experiments with similar results. Statistically significant differences between the treated group were analyzed compared to the control (untreated cells). Intragroup and intergroup comparisons were performed using one-way ANOVA and Tukey’s multiple comparison test. * *p* < 0.05, ** *p* ≤ 0.01, *** *p* ≤ 0.001, n.s. (not statistically significant).

**Figure 6 vaccines-11-00495-f006:**
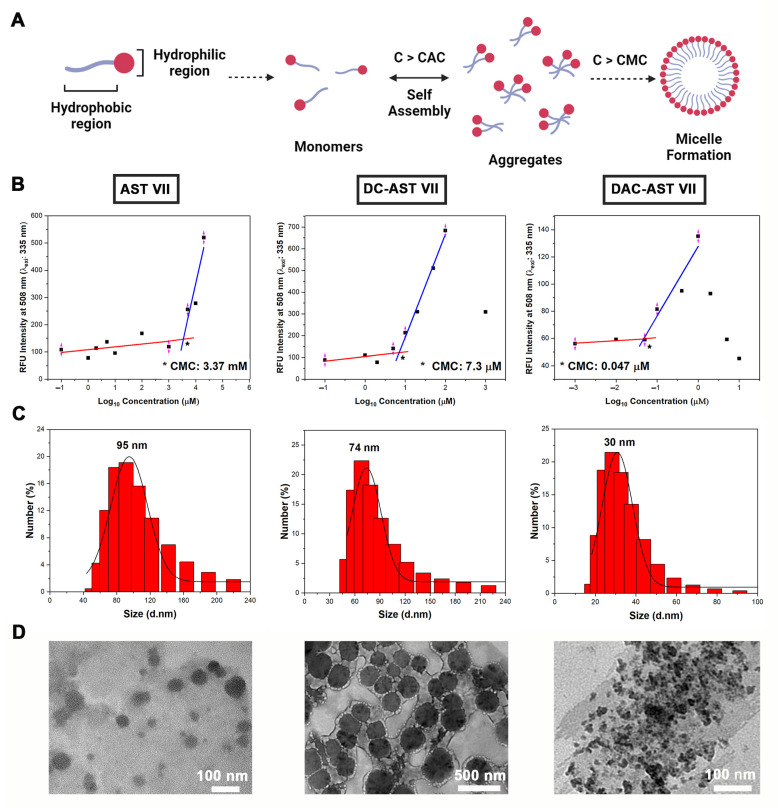
Self-assembling nanoparticles based on *Astragalus* saponins. (**A**) A general illustration for self-assembling particle formation. C: Concentration, CAC: Critical aggregation concentration, CMC: Critical micelle concentration. (**B**) Relative fluorescence intensity at 508 nm vs. logarithm of Astragalus saponins concentrations to determine CMC. (**C**) Representative particle size distribution of self-assembling *Astragalus* saponins that were measured using dynamic light scattering (DLS) analysis. (**D**) Transmission electron microscopy (TEM) images of self-assembled nanoparticles based on AST VII (4000 μM), DC-AST VII (10 μM), and DAC-AST VII (10 μM) in dH_2_O. Data represent two or three independent experiments. Created with BioRender.com. *: the intersection of two linear lines, showing CMC value.

**Figure 7 vaccines-11-00495-f007:**
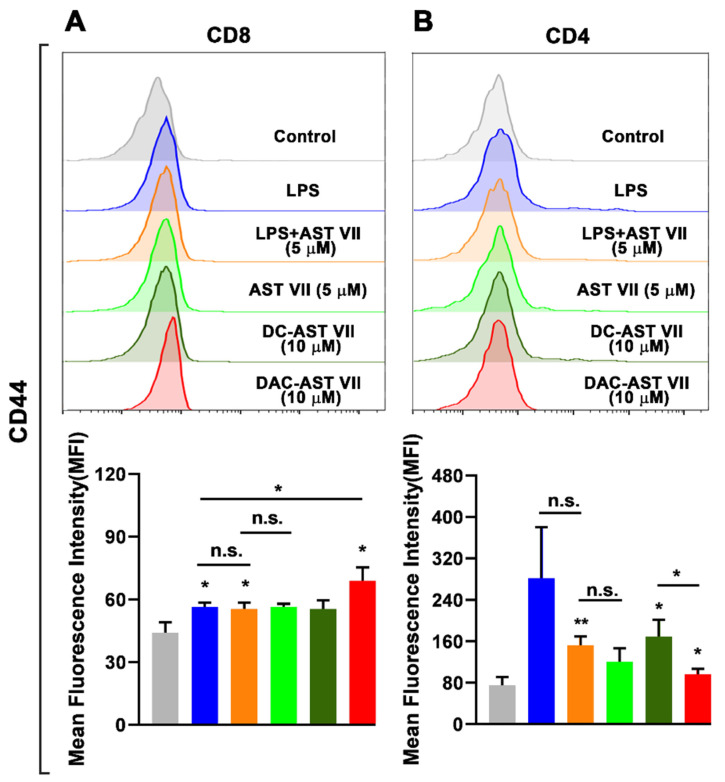
AST VII and its derivatives (DC-AST VII and DAC-AST VII) activated T cells in MLR. BMDCs derived from BALB/c mice were treated with LPS, AST VII, DC-AST VII, or DAC-AST VII for 24 h. The next day, BMDCs were co-cultured with naive CD4^+^ and CD8^+^ T cells isolated from the spleens of C57BL/6 mice for 3 days. The expression of CD44 on (**A**) CD8^+^ T cells and on (**B**) CD4^+^ T cells was determined using flow cytometry. Representative histograms (top panel) and summary graphs (bottom panel) are shown. Data are shown as the mean and SD of triplicates and represent three independent experiments with similar results. Statistically significant differences in the treated group were analyzed compared to the control (untreated cells). Intragroup and intergroup comparisons were performed using one-way ANOVA and Tukey’s multiple comparison test. * *p* < 0.05, ** *p* ≤ 0.01, n.s. (not statistically significant).

## Data Availability

All data are available in the main text or Appendix A.

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
