# Peer review of "Astragalus Saponins, Astragaloside VII and Newly Synthesized Derivatives, Induce Dendritic Cell Maturation and T Cell Activation"

_vaccines, 2023, doi:10.3390/vaccines11030495_

Round 1

Reviewer 1 Report

In this manuscript, Yakubogullari N et al. evaluated the adjuvant potential of AST VII and two AST VII analog derivates in terms of dendritic cell maturation and T cell activation. The experiemnts are well-designed and the obtained results sustain the conclusions raised by the authors. The data presented support the rational design of AST VII (+analogs) as an adjuvant in vaccine formulations for further in vivo evaluation.   

Here are a few minor comments that may benefit the manuscript for publication: 

·      Figure 2. 

o   Panels A and B should be removed from the figure. It is a non-optimized activation experiment that do not contribute to the experimental concussions raised by the authors. 

o   At 1/20 diluted hWB IL-17A secretion should also be quantified (either by ELISA or qPCR)

o   A reduced secretion of TNFa, IL2 and IFNg when cells were treated with QS-21, AST VII and derivates compared to only P+I stimulation is unexpected. Was cell viability checked? This information should be included.

o   P+I stimulation is supraphysiological and conclusions made based on this type of activation may be misleading. Would similar AST VII (+derivates) give a similar cytokine secretion profile under a more physiological T cell activation (CD3/CD28 triggering)? This issue should merit some discussion. 

·      Figure 3,4 and 5. 

o   What is the viability of the BMDMs after the stimulation with AST VII (+derivates)?

·      Figure 7:

o   Despite the T cell activation marker CD44, it would be interesting to evaluate other more sensitive T cell activation markers such as CD69 or CD137 (for CD8 T cells) and OX-40 (for CD4+ T cells). In particular, in this MLR experiment where few differences have been observed among treatment groups and controls, CD44 may not be the most sensible activation marker. 

o   Was there any T cell proliferative difference among the treatment groups? Quantifying CD4 and CD8 T cell proliferation either by a dye dilution assay or simply by T cell counting would help support the conclusions made. 

Reviewer 2 Report

The authors present a follow up report to their 2021 publication in Biologicals. The previous paper demonstrated the efficacy of AST VII as an adjuvant for use with multiple protein antigens.  Furthermore, the current manuscript attempts to elucidate the mechanism of action of AST VII along with 2 other newly developed derivatives. 

The paper is fairly straightforward in design with basic immunologic assays demonstrating how these adjuvants promote dendritic cell and T-cell activation. Although not necessarily definitive the current report provides nice evidence towards the potential in vivo efficacy.

The manuscript will be of interest to a wide audience of those interested in novel immune mechanisms of activation and for those interested in vaccine development. 

I have very few criticisms of the manuscript as written. The below item should be addressed. 

I cannot find any information included on the CD44 antibody used for information in Figure 7. Please add catalog number, clone and fluorophore description in the materials and methods. 
